# Mitigation of Membrane Fouling Using an Electroactive Polyether Sulfone Membrane

**DOI:** 10.3390/membranes10020021

**Published:** 2020-01-30

**Authors:** Chunyan Ma, Chao Yi, Fang Li, Chensi Shen, Zhiwei Wang, Wolfgang Sand, Yanbiao Liu

**Affiliations:** 1Textile Pollution Controlling Engineering Center of Ministry of Environmental Protection, College of Environmental Science and Engineering, Donghua University, Shanghai 201620, China; machunyan@dhu.edu.cn (C.M.); 2181557@mail.dhu.edu.cn (C.Y.); lifang@dhu.edu.cn (F.L.); shencs@dhu.edu.cn (C.S.);; 2Shanghai Institute of Pollution Control and Ecological Security, Shanghai 200092, China; zwwang@tongji.edu.cn; 3State Key Laboratory of Pollution Control and Resource Reuse, School of Environmental Science and Engineering, Tongji University, Shanghai 200092, China; 4Institute of Biosciences, Freiberg University of Mining and Technology, 09599 Freiberg, Germany

**Keywords:** membrane separation, fouling mitigation, low electrical field, electrostatic repulsion

## Abstract

Membrane fouling is the bottleneck limiting the wide application of membrane processes. Herein, we adopted an electroactive polyether sulfone (PES) membrane capable of mitigating fouling by various negatively charged foulants. To evaluate anti-fouling performance and the underlying mechanism of this electroactive PES membrane, three types of model foulants were selected rationally (e.g., bovine serum albumin (BSA) and sodium alginate (SA) as non-migratory foulants, yeast as a proliferative foulant and emulsified oil as a spreadable foulant). Water flux and total organic carbon (TOC) removal efficiency in the filtering process of various foulants were tested under an electric field. Results suggest that under electrochemical assistance, the electroactive PES membrane has an enhanced anti-fouling efficacy. Furthermore, a low electrical field was also effective in mitigating the membrane fouling caused by a mixture of various foulants (containing BSA, SA, yeast and emulsified oil). This result can be attributed to the presence of electrostatic repulsion, which keeps foulants away from the membrane surface. Thereby it hinders the formation of a cake layer and mitigates membrane pore blocking. This work implies that an electrochemical control might provide a promising way to mitigate membrane fouling.

## 1. Introduction

Great expectations are connected with membrane processes to address the challenging issue of water scarcity [1]. Unfortunately, since the birth of membrane water treatment technologies, membrane fouling has been consistently their bottleneck restricting their process efficiency by reduction of water permeation, deterioration of product water quality and an increase of energy consumption [2]. According to the Hermia’s pore blocking models [3], there are four basic types of fouling, i.e., complete blocking, intermediate blocking, standard blocking and cake layer formation. According to the complete blocking model, each molecule that reaches the membrane surface completely blocks the entrance of a membrane pore. The intermediate blocking model is less restrictive, because it considers that some molecules may deposit onto other molecules previously settled. This means that not each molecule, which arrives to the membrane surface, blocks membrane pores. The standard blocking model considers that molecules deposit over the pore walls. As a result, the amount of membrane pores decreases proportionally to the filtered permeate volume. The cake layer formation refers to the formation of a dense layer on the membrane surface. All of them can cause reversible and/or irreversible fouling [4,5,6]. Therefore, it is highly desirable to develop innovative strategies to mitigate fouling of membranes.

Several methods have been proposed to alleviate membrane fouling in other studies [7], such as the pre-treatment of raw water, optimization of operation conditions and membrane cleaning [8,9,10]. However, the methods can alleviate membrane fouling, but it cannot radically solve the problem and consume space and energy. The development of anti-fouling membranes is one of the most promising solutions to this challenging issue [11,12]. Until now, several strategies have been developed to enhance anti-fouling capability of membranes [3,13,14,15]. In particular, the development of electroactive anti-fouling properties is an attractive route [16,17,18]. The technology integrates electrochemistry with membrane separation [19]. The application of a proper electric field has demonstrated to be effective to reduce fouling rates and increase permeate flux in both cross-flow and dead-end filtration systems [20,21,22]. The mechanism of anti-fouling can be explained by different reasons. Both direct effects (direct oxidation/reduction of microbial cells and electrostatic repulsion) and indirect effects (electrochemical H_2_O_2_ production, radical generation, pH and temperature changes and electroosmotic flow) have been suggested as possible anti-fouling mechanisms [23]. For example, Zhang et al. [24] attributed the fouling reduction with the application of a negative potential to the increased energy barrier and the decreased collision efficiency of negatively charged organic matter with the membrane surface. In addition, a microscopic amount of H_2_O_2_ formed electrochemically near the membrane surface could be lethal to bacteria and inhibit biofilm formation during operation. Generally, the membrane materials need to be modified with a conductor to allow for an application of an electric field without deteriorating desirable membrane properties. Numerous bench-scale studies utilize electrostatic repulsion to repel like-charge foulants [25]. Increased electrostatic repulsion between a membrane surface and a foulant can reduce both pore blocking and surface cake formation. For example, Jassby et al. [26] reported that a conductive UF membrane was capable to alleviate significantly the deposition of negatively-charged sodium alginate (SA) with an applied voltage of −3.5 V. In another study Liu et al. [27] developed a robust electroactive thin-film composite (TFC) forward osmosis (FO) membrane, which demonstrated good resistance to SA fouling with the imposition of a 2.0 V DC voltage, if the membrane-electrode assembly was used as a cathode. Despite these improvements, most strategies are only capable of dealing with monotype foulants. However, in the actual sewage are always various types of foulants coexisted. Performance may fail in real conditions for the treatment of industrial wastewater with more complex compositions. Moreover, the synergistic mechanism between the electric field and the membrane needs to be evaluated further for different experimental conditions. In addition, foulant removal efficiency is also another important but often neglected criterion on evaluate membrane performance.

An electroactive membrane operated in cross-flow mode may provide an alternative solution to alleviate the fouling by multiple foulants. The advantages of the cross-flow mode include low membrane fouling, flexible operation and the ability to handle high load feed water quality. Together with electrochemistry, the proposed technology offers the prospect of both improving water flux and enhancing foulant removal efficiency. As a proof-of-concept, we first adopted a classical phase inversion method to fabricate conductive PES membranes. Then, we examined the membrane anti-fouling efficacy under various operational conditions using a few common negatively charged model foulants (e.g., BSA and SA as model non-migratory foulants, yeast as a model proliferative foulant and emulsified oil as a model spreadable foulant). Finally, we propose an underlying mechanism for fouling mitigation based on our experimental evidence. The outcome of this study is providing a robust and promising strategy for alleviating fouling problems from multiple foulants by integrating state-of-the-art electrochemistry and membrane separation.

## 2. Materials and Methods

### 2.1. Chemicals and Materials

*N*-methyl-2-pyrrolodinone (NMP, ≥99.5%), bovine serum albumin (BSA, ≥96%), polyethylene glycol 400 (PEG, Mn = 400g/mol), m-phenylene diamine (MPD, ≥99%), fluorescein isothiocyanate conjugated bovine serum albumin (BSA-FITC) and 1,3,5-benzenetricarbonyl trichloride (TMC, ≥99.9%) were purchased from Sigma-Aldrich (Shanghai, China). Sodium alginate (SA, C_6_H_7_NaO_6_, ≥96%), sodium sulfate (Na_2_SO_4_, ≥96%), sodium dodecyl benzene sulfonate (SDBS, ≥96%), *N*-hexane (≥96%) and sodium chloride (NaCl, ≥96%) were provided by Sinopharm Chemical Reagent Co., Ltd. (Shanghai, China). Polyether sulfone (PES) was provided by BASF Co., Ltd. (Shanghai, China). Edible oil was purchased from Guo Brothers Grain Oil Co., Ltd. (Malaysia, Singapore). Yeast was purchased from Angel Yeast Co., Ltd. (Shanghai, China). Carbon fiber paper was obtained from Ce-Tech Co., Ltd. (Wuhan, China). All solutions were prepared with deionized water (DI-H_2_O) supplied by a Milli-Q Direct 8 purification system (Millipore, Burlington, MA, USA).

### 2.2. Membrane Preparation

A carbon paper-polyether sulfone (Car-PES) ultrafiltration membrane was fabricated via a facile phase inversion method [27,28]. PES powder was mixed with PEG-400, NMP and DI-H_2_O at a ratio of PES:PEG:NMP:DI-H_2_O = 20:37.9:37.9:4.2 (wt%) to prepare a casting dope. Then, highly conductive and porous carbon paper (20 cm × 12 cm × 180 μm) was placed on a glass plate (35 cm × 20 cm) and the dope was cast onto the carbon paper with a casting knife (9 cm × 11 cm × 150 μm). After spreading, the as-casted membrane was immersed in DI-H_2_O together with the glass plate and solidified immediately. After 1 h, the as-casted membrane was immersed in 2 wt% MPD aqueous solution for 60 s. Excess MPD solution was carefully removed from the membrane surface with nitrogen gas. This was followed by contacting the same membrane surface with 0.1 wt% TMC solution (dissolved in *n*-hexane) for 60 s to induce the formation of a polyamide (PA) selective layer on the surface of the PES layer, which formed a dense layer on the membrane surface and improved the membrane selectivity.

### 2.3. Characterizations

Membrane morphology was examined by a HITACH S-480 field-emission scanning electron microscopy (FESEM, HITACHI, Tokyo, Japan). Zeta potential of various foulants solutions were measured by a ZEN 3600 Malvern Zetasizer Nano (Malvern Instruments, Malvern, UK) [29]. Contact angle of membranes was determined by a goniometer (SL-200C, KINO, USA) using DI-H_2_O as a probe liquid. Surface roughness of membranes was examined by atomic force microscopy (FM-nano view 6600 AFM, FEISHIMAN, Suzhou, China). The distribution of proteins on the membrane surface was observed by a fluorescence microscope (ZEISS A1, Carl Zeiss AG, Oberkochen, Germany). Attenuated total reflectance-Fourier transformed infrared spectroscopy (ATR-FTIR, Nicolet 6700, TMO, Waltham, MA, USA) was employed to evaluate the formation of a PA layer on the membrane surface. Total organic carbon (TOC) of the feed and permeate solution was measured by a multi N/C 3100 TOC-VCPH analyzer (Analytik Jena, Jena, Germany). Cyclic voltammetry and anode/cathode potential distribution as a function of total cell potential of the membranes were measured by a CHI760E electrochemical workstation (CH Instruments, Shanghai, China).

The water flux through membranes (J in L/m^2^·h, or LMH) was calculated according to the following Equation:(1)$$J=\frac{\Delta V}{\Delta t\cdot Am},$$
where ΔV(L) is the volume of water permeated through the membrane within Δt (h) and A_m_ is the effective membrane surface area (m^2^).

Selectivity of membranes was measured by the removal efficiency (%) of selected model foulants using the following Equation:(2)$$R=(1-\frac{Cp}{Cf})\times 100\%,$$
where C_p_ and C_f_ (mg/L) are the TOC concentration in the permeate and feed solution, respectively.

### 2.4. Fouling Mitigation Performance

To evaluate the anti-fouling performance of as-fabricated electroactive membranes, three types of model foulants were selected [30]. Non-migratory foulants refer to organic colloids and natural organic matters (NOM), deposit on membrane surfaces and form a rather stable cake layer, we chose BSA and SA as model non-migratory foulants. Proliferative foulants, also referred to as biofoulants, such as bacterial cells, extracellular polymeric substances (EPS) and cell debris, we chose yeast as the model proliferative foulant. Spreadable foulants, including various kinds of oils, not only attach, but also spread, coalesce and form a continuous layer on membrane surfaces, we chose emulsified oil as the model spreadable foulant. The emulsified oil solution was prepared by mixing oil and SDBS in a ratio of 1:9 (wt%). Detailed characteristics of these foulants are summarized in Appendix A.

In a typical filtration process, 10 mg/L of a monotype foulant solution or a mixture of the several foulants was pumped into a self-made cross-flow electrochemical filtration module (see Figure 1). The effects of an applied electric field on fouling mitigation were examined in terms of water flux and foulant removal efficiency. The Car-PES membrane was placed in the middle of the reactor and served as cathode. A perforated Ti sheet with a typical thickness of 2 mm served as an anode. These two electrodes were separated by a silicon gasket. A DC power supply was used to provide the required voltage. Firstly DI-H_2_O at 0.4 MPa for 120 min was infiltrated into the Car-PES membrane. Duration of the filtration experiments was fixed at 60 min at ambient temperature. During filtration a feed solution was pumped into the device at a constant cross-flow rate controlled by a gear pump (WT 3000-1 JA, LONGER, Baoding, China). A constant trans-membrane pressure of 0.1 MPa was maintained. The permeate was collected and monitored by an analytical balance (JA 31002, JINGHAI, Shanghai, China). Unless stated, experiments were performed in triplicate to secure reproducibility. Microscopic analyses were performed at least at three random areas on the membrane surface. Representative results are presented.

## 3. Results and Discussion

### 3.1. Characterization of the Car-PES Membrane

In this work, a carbon paper and the substrate polymer form an interpenetrating mixture [27]. Figure 2 compares the FESEM images of a pristine PES membrane and a Car-PES membrane. The structure of the Car-PES membrane has three layers, the PA layer on the top, the PES layer in the middle and the carbon paper on the bottom. The thickness of the Car-PES membrane is 224.2 ± 7.6 μm, 1.2 times thicker than the PES membrane with 188.9 ± 2.5 μm. The as-fabricated Car-PES membrane is conductive only on the carbon paper side, while the polymeric side is non-conductive (Appendix A). The PES membrane showed characteristic finger-like cavities with a thin selective layer (10 μm thickness) formed ontop of the PES membrane surface (Figure 2b). For the Car-PES membrane, the presence of paper disrupted the formation of finger-like porous structures in the PES membrane (Figure 2d). Instead, a thin and sponge-like structure was formed on top of the Car-PES membrane surface [31]. Moreover, the presence of carbon paper support created a vastly different bottom layer. As can be seen from Figure 2c, the pores in the carbon paper were not completely occupied by the PES [27], and the Car-PES membrane led a decreased pore size compared with the PES (D = 65.1 nm vs. D = 77.0 nm), which was obtained by the filtration velocity method (Appendix A). As shown from the Figure 2e,f, as expected, the morphology of the top layer of both membranes were similar, due to the absence of carbon fibers in this layer.

The permeability of membrane materials is closely related to its morphology, porosity and wetability [32]. Thus, we compared the difference in roughness between a Car-PES membrane and a PES membrane using the AFM technique. As shown in Figure 3a,b, the mean root squared (Rq) surface roughness of the Car-PES and the PES membrane without the PA layer was determined to be 77.1 ± 4.8 nm and 73.5 ± 2.6 nm, respectively. It is reasonable to expect that the carbon paper underneath the PES layer posed a negligible effect on the membrane surface. Figure 3c,d indicated the formation of a PA layer, which reduced the surface roughness for both the Car-PES (41.3 ± 3.6 nm) and the PES (40.1 ± 3.2 nm). The decrease in roughness could be attributed to the interfacial polymerization on the Car-PES and PES membrane surface, leading to the formation of a dense film ontop [33]. Moreover, the contact angle measurements were conducted to evaluate the wettability of both membranes. As is obvious in Appendix A, the contact angle on the top surface of the Car-PES membrane was only slightly higher than that of PES membrane, despite the hydrophobic nature of carbon paper. This indicates that the hydrophobic carbon paper might be encapsuled in the infiltrated PES polymer.

The decline in water flux indirectly reflects the extent of fouling of the membranes. Therefore, we measured the change of water flux, while challenging two kinds of membranes with selected model foulants. As shown in Figure 4, the water flux of the Car-PES membrane was always lower than that of the PES counterpart regardles of the foulants. One possible reason for such a decrement in water flux through the Car-PES membrane is possibly the formation of a thin-layer of a sponge-like structure on the Car-PES membrane surface. Such high density sponge-like structures exert additional resistance to water permeability [31]. However, since the thickness of this sponge-like structure is only 1–2 μm, its negative impact on the water flux of the Car-PES membrane should be limited. When filtering selected foulants, both the PES and the Car-PES membranes, demonstrated a similar trend. Under the same conditions, the membrane fouling caused by yeast and oil was more serious, than by SA and BSA. For both membrane materials, the water flux declined to 30% ± 5%, 33% ± 3%, 20% ± 3% and 21% ± 2%, respectively, over 1 h continuous filtration of BSA, SA, yeast and emulsified oil. This indicates that the introduction of a carbon paper support had a negligible impact on the water flux in the absence of an electric field [34]. Over 24 h continuous filtration, the flux changing trend was similar with that of 1 h (Appendix A). Due to the electric insulative nature of PES alone, the anti-fouling performance of the PES membrane was not affected by the electric field. The one-hour anti-fouling experiment of PES membrane also proved to this point (Appendix A). Further experiments mainly focused on the electroactive Car-PES membrane with special focus on the boosting effect of the electric field on the fouling mitigation.

### 3.2. Effect of Ionic Strength on Membrane Fouling

Extensive studies indicate that an increased ionic strength is favorable for fouling mitigation under electrochemical assistance [24,35]. Herein, upon application of an applied voltage of −1.0 V on the Car-PES membrane, various Na_2_SO_4_ concentrations (from 0 to 20 mM) were spiked into 10 mg/L SA solution for challenging the Car-PES membrane. As presented in Figure 5, the water flux of the control assay, decreased by 77.3% within a 1 h filtration process. Compared with the control (without Na_2_SO_4_), the water flux through the Car-PES membrane was improved during 1 h only by 4.0% with a Na_2_SO_4_ concentration of 5 mM. If the Na_2_SO_4_ concentration was increated to 10 mM, the water flux increased by 27.0% over 1 h compared to the control. This result suggests that an increased ionic strength strengthens the electric field effect of the electroactive Car-PES membrane against charged foulants. Thus, it is the mitigating membrane fouling to a certain extent [24]. The primary fouling mitigation mechanism was the potential-induced cathodic surface charge (Appendix A), which increases the Derjaguin–Landau–Verwey–Overbeek (DLVO) energy barrier and decreases the collision efficiency of negatively charged organic compounds with the membrane surface [24]. Further increasing the Na_2_SO_4_ concentration to 15 or 20 mM contributes negatively to the water flux of the Car-PES membrane. Bowen et al. have shown that an excessive increase of the ionic strength may cause a reduction of the water flux by a compression of the electric double-layer [36]. Therefore, 10 mM of Na_2_SO_4_ were identified as optimal and used for the subsequent experiments.

### 3.3. Effect of the Electric Field on Mitigation of Membrane Fouling

Various investigations suggest that electrostatic interactions may play an important role in the performance of anti-fouling membranes [37,38,39,40]. Application of a cathodic current is known to promote the detachment of foulants from the electrode surface as a result of the electrostatic and electrophoretic repulsive forces generated on the membrane surface [40]. In this study, we selected negatively charged model compounds as model foulants for the proof-of-concept (Appendix A). To evaluate the anti-fouling and separation performance of a Car-PES membrane, the change of water fluxes while filtering different foulants were recorded under different applied voltages (0, −0.5, −1.0, −2.0 and −3.0 V). As shown in Figure 6a, during the filtration of 10 mg/L BSA together with 10 mM Na_2_SO_4_ solutions the water flux decreased by 70% within 1 h in the absence of an electric field. This can be due to the convective deposition of protein molecules onto the Car-PES membrane surface [41]. Further promoting the deposition of protein molecules and exacerbating the membrane fouling with continuous operation. However, the membrane fouling caused by BSA was suppressed mostly in the presence of a negative voltage. The water flux was 0.43, 0.48, 0.66 and 0.70 after operating for 1 h in the Car-PES membrane at −0.5, −1.0, −2.0 and −3.0 V, respectively. This caused a flux being 1.4, 1.6, 2.2 and 2.3 times as high as in the absence of an electric field (0.30). The increased water flux suggests that the fouling of the Car-PES membrane could be partially mitigated by applying an electric field. The measured currents in the experiments were less than <0.15 A m^−2^ (Appendix A), indicating Faradaic electron transfer reactions were negligible as expected from the Car-PES membrane. Thus, only electrostatic repulsion effects will be considered in regards to the anti-fouling mechanism. Due to BSA being electronegative, electrostatic repulsion occurred on the Car-PES membrane surface under electric field, thus pushing the BSA molecules away from the membrane surface and reducing membrane fouling. Fluorescence images of the Car-PES membrane taken after 1 h continuous operation confirmed that the application of an electric field alleviated membrane fouling evidently (Appendix A). The strong fluorescence signal shows that, without an electric field, the pristine Car-PES membrane was covered completly by BSA molecules. Since carbon paper does not show any affinity to BSA molecules, the strong fluorescence exhibited by the Car-PES membrane is another indication of the encapsulation of the carbon fibers by the PES polymer. However, if the applied voltage is increased from 0 to −3.0 V, the adsorption of the BSA was greatly suppressed and the fluorescence became gradually reduced. With the increase of the applied voltage, the amount of deposited BSA molecules tended to be pushed away from the membrane surface by electrostatic repulsion. These promising data suggest that an electroactive membrane design enables fouling mitigation by negative-charged BSA molecules.

The decline in water flux through the Car-PES membrane was very obvious, if challenging it with 10 mg/L SA together with 10 mM Na_2_SO_4_ solution (by 70.9%) over 1 h without application of an electric field. Once a negative voltage was applied, the fouling extent of the Car-PES membrane was evidently mitigated due to the like-charge repulsion between the alginate anions and the electric field. SA could easily ionize in feed solution to provide an abundance of alginate anions, generating the like-charge repulsion between the alginate anions and the applied voltage, thus slowing the decline in water flux. As can be seen from the Appendix A, the Car-PES membrane surface charge changed in the presence of SA foulants and negative electric filed simultaneously. When the applied voltage reached −3.0 V, the membrane surface charge reached −1.48 V vs. Ag/AgCl. It is reasonable to expect that the stronger interaction force formed between the Car-PES membrane surface and the SA foulants, causing high permeability flux. For example, the water flux of the SA solution was 0.74 at −3.0 V, which was 45% higher than the value, if no voltage was applied. The increment of 45% is slightly higher than that for BSA, where the water flux was increased by 40% under similar conditions. Perhaps the higher electronegativity (−26.6 mV) and the larger particle size (346.2 ± 34.7 nm) of SA than for BSA (−11.6 mV, 5.4 ± 0.6 nm), which resulted in a greater electrostatic repulsion force caused that effect. The presence of an electrostatic repulsion force pushing SA molecules away from the membrane surface, thereby hindering the formation of a gel layer, mitigating membrane pore blocking in the Car-PES and then alleviating flux loss.

As expected, the worst case was obtained when challenging a solution of 10 mg/L yeast together with 10 mM Na_2_SO_4_, due to the large yeast particle size (3.3 ± 0.4 μm). The initial water flux was reduced by 80% within 1 h in the presence of the yeast cake. As shown in Figure 6c, if an electric field of −0.5 V was applied, the flux of the Car-PES membrane changed negligibly compared with the value in the absence of an electric field. With increasing the applied voltage from −1.0 to −3.0 V, the water flux increased from 0.38 to 0.76 over 1 h of continuous operation. This value is 3.8 times as high as that in the absence of an electric field. Anti-fouling mechanism of yeast foulant can be attributed to the electrostatic repulsion influence of negative electric field. The negative electric field was applied to the Car-PES membrane surface, electrostatic and electrophoretic forces (vertical) dominated the yeast detachment. The anti-fouling experiment indicates that sufficiently high (−0.5 to −3V) electrical potentials can prevent the attachment of yeast to an electrically charged membrane surface after the initial deposition step.

The spreadable oil foulants are unstable compared to other foulants owing to their easy deformation and coalescence in an aqueous solution [42]. The variation tendency of the water flux for the Car-PES membrane is not evident, if challenged by a solution cointaining 10 mg/L emulsified oil together with 10 mM Na_2_SO_4_ under a negative electric field. Oil foulants are easily deposited on membrane surfaces due to their strong hydrophobicity by hydrophobic interactions between oil and membrane materials [43]. If oil droplets touch the underwater membrane surface, they deform, infiltrate and spread out along the membrane–oil–water three-phase interfaces [28]. As can be seen in Figure 6d, the water flux through a Car-PES membrane increased only by 28% in the presence of an electric field within 1 h, compared with no voltage applied (from 0.21 at 0 V to 0.49 at −3.0 V). We, thus, extrapolated that electrostatic repulsion of the oil from the charged surface of the Car-PES membrane was not strong enough to repel the oil. As can be seen from the Appendix A, the change of membrane surface charge was not obvious in the presence of emulsified oil foulants and negative electric filed. When the applied voltage reached −3.0 V, the membrane surface charge reached −1.16 V vs. Ag/AgCl. This led to the smaller interaction force between the Car-PES membrane surface and the emulsified oil foulants, causing severe membrane fouling and rapidly decreasing permeate flux compared with other types of foulants. In addition, Appendix A showed that the electronegativity of emulsified oil was high (66.4 mV and 13.9 mV), but by its particle size analysis we could find that most particle size of emulsified oil used in this study were at the micron level (13.6 μm). It is reasonable to expect that the own gravity of emulsified oil is bigger than applied electric field force and the effect of mitigating membrane fouling was not as obvious as other types of foulants.

It is worth noting that some positive ions might adsorb on the membrane and make the Car-PES membrane surface electropositive [44], as suggested by Appendix A. However, experimental results show that within the scope of this study, the application of −3.0 V was sufficient to mitigate membrane fouling even in the presence of the positive ions selected. Further increasing voltage to −4.0 V contribute only negligibly to the improvement of the water flux (e.g., <0.5%). In addition, if the applied voltage reaches −4.0 V, water electrolysis can occur on the membrane surface, which is leading to a cousiderably decreased current efficiency [45]. The gas bubbles may also block the pores size of the membrane, further interfering with the water flux.

Meanwhile, the changes of the TOC removal efficiency for various foulants under different negative voltages are summarized in Figure 7. Feed samples were collected at the start and at the end of each experiment. Permeate samples were taken only at the end of each experiment. In the absence of an electrical field, the TOC removal efficiency for 10 mg/L BSA together with 10 mM Na_2_SO_4_ solution was only 33%. In this work, the hydrodynamic diameter of BSA (5.4 ± 0.6 nm) is smaller than the Car-PES membrane pore size (65.1 ± 5.8 nm). It indicates that BSA it can easily pass through the membrane pore in the absence of an electric field. If the applied voltage was increased from −0.5 to −3.0 V, also the TOC removal efficiency for BSA gradually increased from 54% to 91%. The TOC removal efficiencies for the assay with 10 mg/L SA were 92%, 93%, 94% and 97% at −0.5, −1.0, −2.0 and −3.0 V, respectively. It means that values were 2.1, 2.2, 2.2 and 2.3 times as high as those without an electric field (42%). In this work, the hydrodynamic diameter of BSA (5.4 ± 0.6 nm) was smaller than the Car-PES membrane pore size (65.1 ± 5.8 nm). It indicates that BSA molecules could easily pass through the membrane pore in the absence of electric field. The TOC removal of SA suggest that SA molecules tend to be ionized in solution to produce alginate anions with smaller size, so that it can pass through the membrane pores with water flow and deteriorate the SA removal efficiency. In the presence of an electric field, the removal efficiency of SA was increased due to electrostatic repulsion between alginate anions and the negative electric field. The TOC removal efficiency for 10 mg/L yeast solution remained at a high level (>90%), since the particle size (micron-level) of yeast is much larger than the pore size of the Car-PES membrane (nano-level). Therefore, the yeast cell can hardly pass through the Car-PES membrane. Oil can be easily deformed, diffused and adhered in aqueous solution. Thus, it is unstable in feed solution and the part of oil molecules may adhere to water molecules and pass through membrane pores. As the voltage increased, the electrostatic repulsion between the oil molecules and the electric field increased. As shown in Figure 7d, the TOC removal efficiency for 10 mg/L emulsified oil together with 10 mM Na_2_SO_4_ solution at −3 V (94%) was 1.5 times higher than that without an electric field (62%). Appendix A presents cyclic voltammetry data for the Car-PES membrane in the presence of the four selected foulants. It shows that no redox peaks appeared. Obviously the fouling mitigation is not caused by redox reactions. Meanwhile, Appendix A shows that the maximum current is less than 1.5 mA during the whole filtration process, indicating that the Car-PES membrane is a non-faraday cathode and the current generated in this experiment is a non-faraday current. Therefore, the strong electrostatic repulsion between foulants and membrane surface play a dominant role in the fouling mitigation. The enhancement of the TOC removal efficiency by hindering the transmembrane transport of the negativly charged foulants is the result.

To further evaluate the effect of applied voltage on mitigation of the membrane fouling, SEM images of the Car-PES membrane after separating yeast at different voltages are exhibited in Figure 8. Figure 8a, b shows that a dense yeast layer was attached to the membrane surface over 1 h of continuous operation in the absence of an electric field. It means that the Car-PES membrane was fouled by yeast cells. No evident difference was visible for the fouling extent at −0.5 V, compared with that in the absence of the electric field, which is consistent with the change of water flux. This finding can be associated with the Car-PES membrane surface bias potential (+0.2 V vs. Ag/AgCl) shown in Appendix A, indicating that the surface of the Car-PES membrane is electropositive. The electrostatic attraction between the positively charged membrane surface and negatively charged yeast cells (−11.2 mV) lead to cell adsorption and caused severe membrane fouling. If an increased voltage was applied to the Car-PES membrane surface (e.g., increased from −1.0 to −3.0 V), only a few yeast cells were attached to the membrane surface. Electrostatic repulsion [46] between negatively charged yeast cells and the electroactive membrane, resulted in yeast being swept away by cross flow into feed solution and mitigated to some extent yeast adhesion onto the Car-PES membrane surface.

### 3.4. Fouling Mitigation with a Mixture of Various Foulants

Based on the above results, an electric field was applied to prove its effect on fouling mitigation of separate types of negatively charged foulants. Since these types of foulants may co-exist in environmental water matrix [47], comprehensive fouling mitigation of the Car-PES membrane was evaluated by challenging it with a mixture of the several foulants (10 mg/L BSA, 10 mg/L SA, 10 mg/L yeast, 10 mg/L emulsified oil and 10 mM Na_2_SO_4_) under different voltages. The changes of water flux and TOC removal efficiency by the Car-PES membrane as a function of applied voltage were measured and are summarized in Figure 9. At an applied voltage of −0.5, -1.0, −2.0 and −3.0 V, the water fluxes through the Car-PES were 0.37, 0.47, 0.58 and 0.67 over 1 h continuous operation. These values were 1.4, 1.8, 2.2 and 2.6 times higher than those in the absence of an electric field (0.26), respectively. Electrostatic repulsion forces between the negatively charged foulants and the catholically polarized Car-PES membrane mitigate the membrane fouling to a certain extent, as is indicated by the increase in flux. The TOC removal efficiency can be improved in the presence of an electric field. It increased from 65% at 0 V to 90.7% at −3.0 V. This confirms that fouling mitigation with a mixture of various foulants is effective. Furthermore, the water flux and TOC removal efficiency of the Car-PES membrane challenged by a mixture of BSA, SA, yeast and emulsified oil, are good for a single foulant. In addition to electrostatic repulsion, the interactions among different categories of foulants may also contribute to fouling mitigation. A previous study reported that protein–yeast interactions led to a loose fouling layer with decreased specific resistance. However, this mixture was still able to pore block by capturing a significant fraction of the protein aggregates [48]. Furthermore [49], in a previous report the interactions between proteins and polysaccharides have been used to mitigate emulsified oil fouling on membrane surfaces.

## 4. Conclusions

In summary, an electroactive PES membrane with excellent anti-fouling properties by addition of electrostatic repulsion into the filtration process was designed. Such efficacy was demonstrated by treating various types of foulants (e.g., BSA, SA yeast and emulsified oil) as well as a mixture of these foulants under electro-assistance. The strong electrostatic repulsion of the membrane surface would hinder the transmembrane transport of the negative charged foulants that could mitigate multiple membrane fouling and reduce flux loss. When compared with that in the absence of an electric field, the water fluxes were increased by 40%, 45%, 56% and 28%, respectively, while passing through solutions of BSA, SA, yeast and emulsified oil at −3 V. Meanwhile, the TOC removal efficiencies were improved by 91%, 97%, 90% and 94% correspondingly. Furthermore, the membrane presented high anti-fouling abilities for removing a mixture of various foulants (containing BSA, SA yeast and oil), resulting in an increased flux 2.6 times as high as without an electric field. This work indicates that electroactive PES membranes may provide a new strategy for designing next-generation anti-fouling membranes for wastewater treatment.

## Figures and Tables

**Figure 1 membranes-10-00021-f001:**
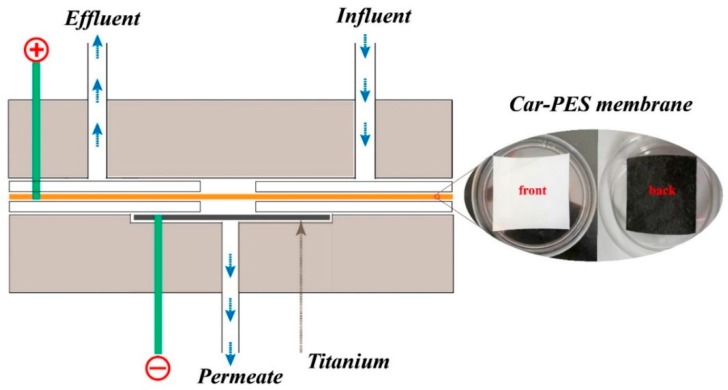
Electrochemical cross flow filtration model of Car-PES composite conductive separation membrane.

**Figure 2 membranes-10-00021-f002:**
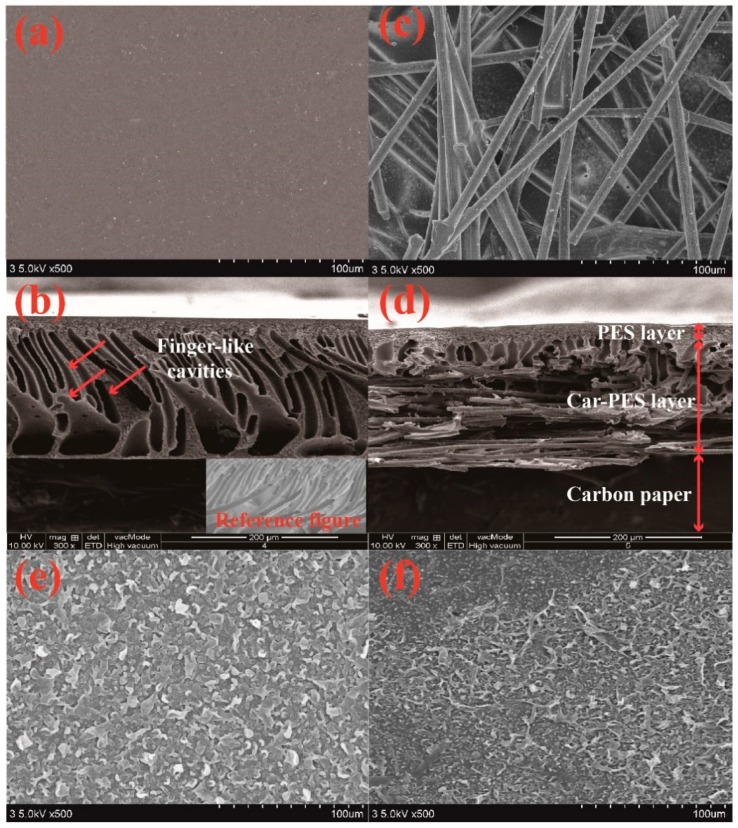
SEM images of the bottom, cross-sections and top of PES (**a**,**b**,**e**) and Car-PES (**c**,**d**,**f**) membranes: (**a**,**c**) bottom surfaces, (**b**,**d**) cross-sections and (**e**,**f**) top surfaces.

**Figure 3 membranes-10-00021-f003:**
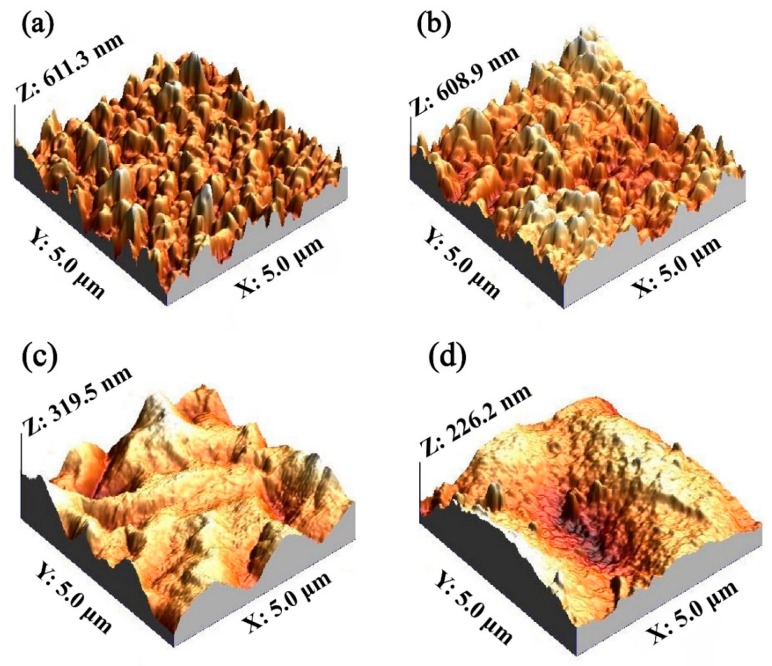
Surface roughness: (**a**) PES membrane without a PA layer, (**b**) Car-PES membrane without a PA layer, (**c**) PES membrane with a PA layer and (**d**) Car-PES membrane with a PA layer.

**Figure 4 membranes-10-00021-f004:**
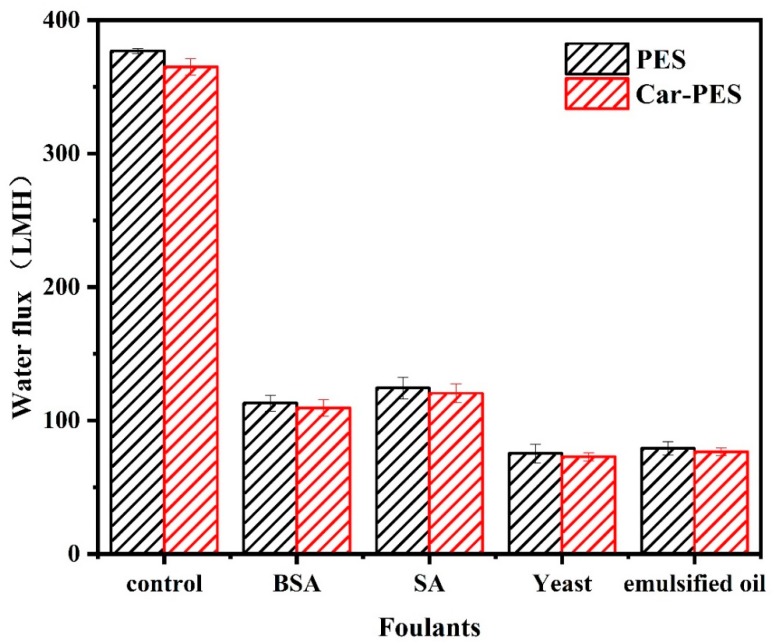
Water fluxes through PES and Car-PES membranes (conditions: (BSA)_in_ = 10 mg/L, (SA)_in_ = 10 mg/L, (yeast)_in_ = 10 mg/L, (emulsified oil)_in_ = 10 mg/L, (cross-flow velocity)= 6.1 cm/s, (time) = 60 min and (pressure) = 0.1 bar).

**Figure 5 membranes-10-00021-f005:**
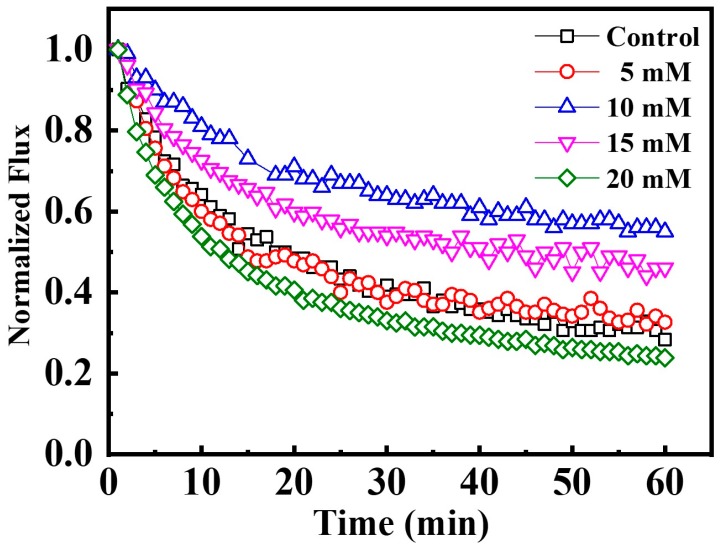
Anti-fouling performance as a function of ionic strength. Car-PES cathode and Ti anode: the 1 h normalized permeability vs. aqueous Na_2_SO_4_ concentration; (conditions: (SA)_in_ = 10 mg/L and (voltage) = −1.0 V).

**Figure 6 membranes-10-00021-f006:**
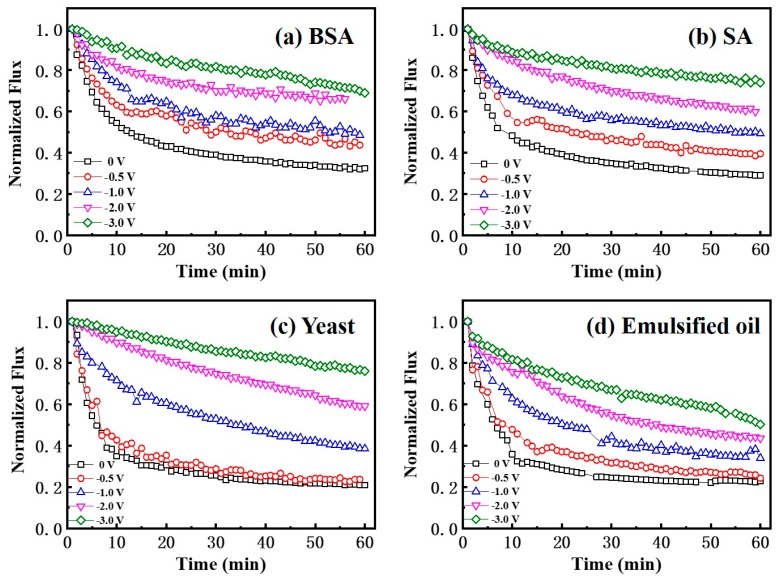
Normalized flux through a Car-PES membrane in comprehensive anti-fouling experiments with different foulant solutions and various negative voltages applied: (**a**) BSA, (**b**) SA, (**c**) yeast and (**d**) emulsified oil (conditions: (BSA)_in_ = 10 mg/L, (SA)_in_ = 10 mg/L, (yeast)_in_ = 10 mg/L, (emulsified oil)_in_ = 10 mg/L, (Na_2_SO_4_) = 10 mM, (cross-flow velocity) = 6.1 cm/s and (pressure) = 0.1 bar.)

**Figure 7 membranes-10-00021-f007:**
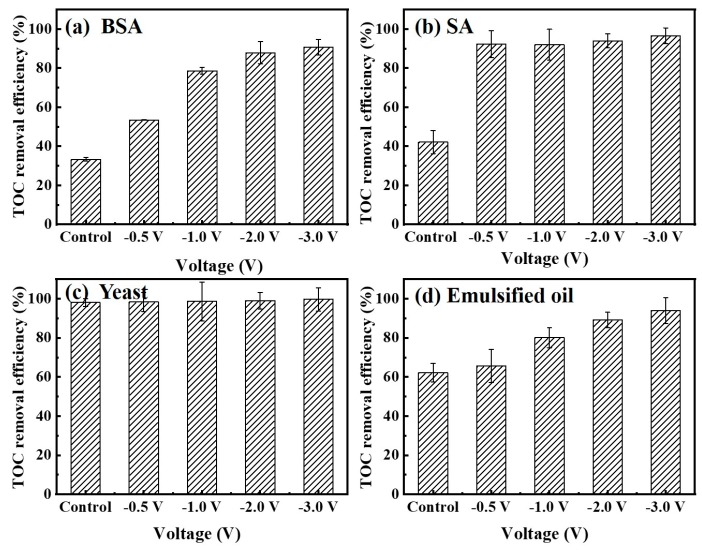
Total organic carbon (TOC) removal efficiency by a Car-PES membrane demonstrating anti-fouling capacity with various foulant solutions under different negative voltages: (**a**) BSA, (**b**) SA, (**c**) yeast and (**d**) emulsified oil (conditions: (BSA)_in_ = 10 mg/L, (SA)_in_ = 10 mg/L, (yeast)_in_ = 10 mg/L and (emulsified oil)_in_ = 10 mg/L).

**Figure 8 membranes-10-00021-f008:**
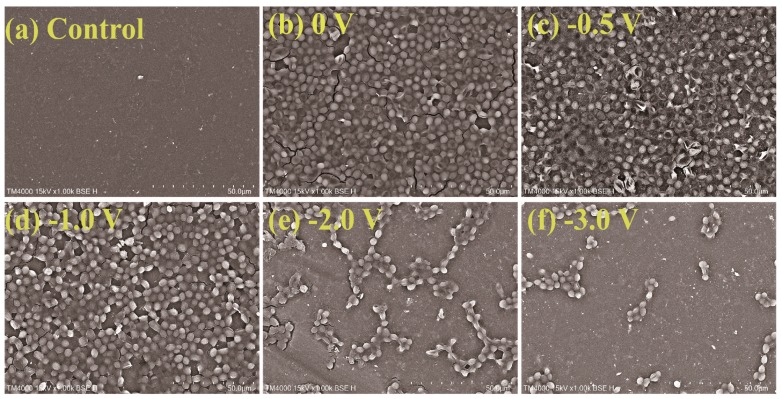
Adhesion of yeast cells on membrane surfaces under different voltage conditions after 1 h of filtration: (**a**) control, (**b**) 0 V, (**c**) −0.5 V, (**d**) −1.0 V, (**e**) −2.0 V and (**f**) −3.0 V.

**Figure 9 membranes-10-00021-f009:**
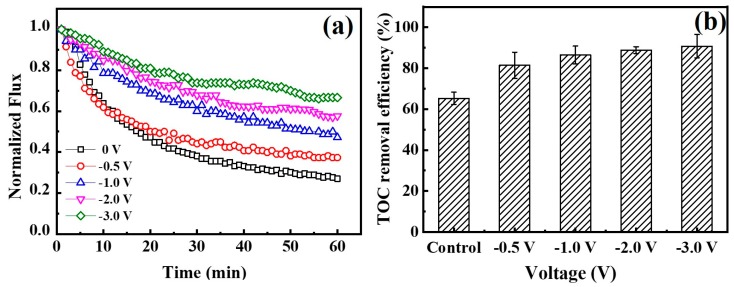
(**a**) Normalized fluxes through a Car-PES membrane at different voltages applied. (**b**) TOC removal efficiency for a Car-PES membrane at different voltages (conditions: (BSA)_in_ = 10 mg/L, (SA)_in_ = 10 mg/L, (yeast)_in_ = 10 mg/L, (emulsified oil)_in_ = 10 mg/L, (Na_2_SO_4_) = 10 mM, (cross-flow velocity) = 6.1 cm/s and (pressure) = 0.1 bar).

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
