# Peer review of "Mitigation of Membrane Fouling Using an Electroactive Polyether Sulfone Membrane"

_membranes, 2020, doi:10.3390/membranes10020021_

Round 1

Reviewer 1 Report

Well-written and interesting article. Only a few text-editing corrections should be made:

Line 218: caused

Line 236: membrane

Author Response

Comments by Reviewer #1:

Well-written and interesting article. Only a few text-editing corrections should be made.

Line 218: caused

Reply: We appreciate the reviewer’s positive comments. We have corrected this mistake.

Revision:

Line 258,

“However, the membrane fouling caused by BSA was suppressed mostly in the presence of a negative voltage.”

Line 236: membrane.

Reply: We have corrected this mistake.

Revision:

Line 281,

“Once a negative voltage was applied, the fouling extent of the Car-PES membrane was evidently mitigated due to the like-charge repulsion between the alginate anions and the electric field.”

Reviewer 2 Report

In the manuscript, electroactive polyethersulfone (PES) membranes capable of mitigating fouling were fabricated by casting PES dope solution on carbon paper. Three types of foulants, including non-migratory foulant (bovine serum albumin & sodium alginate), proliferative foulant (yeast) and spreadable foulant (emulsified oil), were used to evaluate the anti-fouling properties of the prepared membranes. By applied a higher voltage across the membrane, the membrane had better anti-fouling efficacy not only to all the single foulant but also to a mixture of foulants. However, the use of carbon paper to improve anti-fouling performance of electroactive membranes have been studied (reference 19). The current manuscript does not show a significant difference in experimental design. The authors have to emphasize the novelty of their study. Furthermore, some concepts and data interpretations need to be clarified to make the manuscript easy to understand. The manuscript is not suitable for publication in Membranes at the current status. Rejection is suggested.

Some specific comments are listed below:

1.       Only four typical foulant models are not able to represent wide-spetrum foulants. Please revise the title.

2.       Section 2.2: what is the thickness of the carbon paper?

3.       Section 2.4: “Duration of the filtration experiments was fixed at 60 min at ambient temperature.” Was the filtration tested for a long time? How was 1 h chosen for the experiment?

4.       Section 3.1: “pore size compared with the PES (D = 65.1 nm vs. D =77.0 nm).” How was the membrane pore size obtained?

5.       The pore sizes are much larger than typical thin-film composite (TFC) membranes prepared with MPD and TMC. What is the purpose of the TFC layer? Why the rejections to bovine serum albumin & sodium alginate are so low?

6.       “The decrement of the surface roughness of the Car-PES membrane can be explained by the coverage of the carbon paper on the PA layer” why is the membrane roughness lower than the membrane without carbon paper? What is the structure of the membrane (PA layer on the top)? Please also provide the FESEM images of the membrane top surfaces.

7.       “As is obvious in Fig. S1, the contact angle on the back surface of the Car-PES membrane was only slightly higher than that of PES membrane” why was contact angle on membrane back surface tested? Please also provide the contact angles of membrane top surfaces.

8.       Figure 4. What was the original flux of car-PES and the PES membranes? Why the unit of normalized flux is LMH?

9.       Figure 5. The authors only tested the auti-fouling performance of car-PES. How was the fouling performance of the PES membrane? Was the performance similar to car-PES sample? Will charge influence the anti-fouling performance of the PES membrane? Please justify.

10.   Figure 7. Why do membranes without applied voltage have lower TOC removal efficiencies? Please justify.

Author Response

Comments by Reviewer #2:

In the manuscript, electroactive polyethersulfone (PES) membranes capable of mitigating fouling were fabricated by casting PES dope solution on carbon paper. Three types of foulants, including non-migratory foulant (bovine serum albumin & sodium alginate), proliferative foulant (yeast) and spreadable foulant (emulsified oil), were used to evaluate the anti-fouling properties of the prepared membranes. By applied a higher voltage across the membrane, the membrane had better anti-fouling efficacy not only to all the single foulant but also to a mixture of foulants. However, the use of carbon paper to improve anti-fouling performance of electroactive membranes have been studied (reference 19). The current manuscript does not show a significant difference in experimental design. The authors have to emphasize the novelty of their study. Furthermore, some concepts and data interpretations need to be clarified to make the manuscript easy to understand. The manuscript is not suitable for publication in Membranes at the current status. Rejection is suggested.

Reply: Based on the reviewer’s comments, we have reorganized the manuscript structure to emphasize the novelty, provided more detailed discussion on the mechanism section, and polished the language to improve the readability. Thank you for all these good suggestions.

Revision:

Line 17,

“Membrane fouling is the bottleneck limiting the wide application of membrane processes. Herein, we adopted an electroactive polyether sulfone (PES) membrane capable of mitigating fouling by various negatively-charged foulants. To evaluate anti-fouling performance and the underlying mechanism of this electroactive PES membrane, three types of model foulants were selected rationally (e.g., bovine serum albumin (BSA) and sodium alginate (SA) as non-migratory foulants, yeast as proliferative foulant and emulsified oil as spreadable foulant). Water flux and TOC removal efficiency in the filtering process of various foulants were tested under an electric field. Results suggest that under electrochemical assistance, the electroactive PES membrane has an enhanced anti-fouling efficacy. Furthermore, a low electrical field was also effective in mitigating the membrane fouling caused by a mixture of various foulants (containing BSA, SA, yeast and emulsified oil). This result can be attributed to the presence of electrostatic repulsion, which keeps foulants away from the membrane surface. Thereby it hinders the formation of a cake layer and mitigates membrane pore blocking. This work implies that an electrochemical control might provide a promising way to mitigate membrane fouling.”

Line 58,

“The mechanism of anti-fouling can be explained by different reasons. Both direct effects (direct oxidation/reduction of microbial cells and electrostatic repulsion) and indirect effects (electrochemical H2O2 production, radical generation, pH and temperature changes, electroosmotic flow) have been suggested as possible anti-fouling mechanisms [23]. For example, Zhang et al. [24] attributed the fouling reduction with the application of a negative potential to the increased energy barrier and the decreased collision efficiency of negatively charged organic matter with the membrane surface. In addition, a microscopic amount of H2O2 formed electrochemically near the membrane surface could be lethal to bacteria and inhibit biofilm formation during operation [25].”

Line 74,

“Despite these improvements, most strategies are only capable of dealing with mono-type foulants. However, in the actual sewage are always various types of foulants coexisted. Performance may fail in real conditions for the treatment of industrial wastewater with more complex compositions. Moreover, the synergistic mechanism between the electric field and the membrane needs to be evaluated further for different experimental conditions. In addition, foulant removal efficiency is also another important but often neglected criterion on evaluate membrane performance.

An electroactive membrane operated in cross-flow mode may provide an alternative solution to alleviate the fouling by multiple foulants. The advantages of cross-flow mode include low membrane fouling, flexible operation and the ability to handle high load feed water quality. Together with electrochemistry, the proposed technology offers the prospect of both improving water flux and enhancing foulant removal efficiency. As a proof-of-concept, we first adopted a classical phase inversion method to fabricate conductive PES membranes. Then, we examined the membrane anti-fouling efficacy under various operational conditions using a few common negatively-charged model foulants (e.g., BSA and SA as model non-migratory foulants, yeast as model proliferative foulant, emulsified oil as model spreadable foulant). Finally, we propose an underlying mechanism for fouling mitigation based on our experimental evidence. The outcome of this study is providing a robust and promising strategy for alleviating fouling problems from multiple foulants by integrating state-of-the-art electrochemistry and membrane separation.”

Line 262,

“The measured currents in the experiments were less than <0.15 Ažm−2 (Fig. S8), indicating Faradaic electron transfer reactions were negligible as expected from the Car-PES membrane. Thus, only electrostatic repulsion effects will be considered in regards to the anti-fouling mechanism. Due to BSA is electronegative, electrostatic repulsion occurred on the Car-PES membrane surface under electric field, thus pushing the BSA molecules away from the membrane surface and reducing membrane fouling.”

Line 283,

“SA could easily ionize in feed solution to provide an abundance of alginate anions, generating the like-charge repulsion between the alginate anions and the applied voltage, thus slowing the decline in water flux. As can be seen from the Fig. S7b, the Car-PES membrane surface charge changed in the presence of SA foulants and negative electric filed simultaneously. When the applied voltage reached -3.0 V, the membrane surface charge reached -1.48 V vs. Ag/AgCl. It is reasonable to expect that the stronger interaction force formed between the Car-PES membrane surface and the SA foulants, causing high permeability flux.”

Line 303,

“Anti-fouling mechanism of yeast foulant can be attributed to the electrostatic repulsion influence of negative electric field. The negative electric field was applied to the Car-PES membrane surface, electrostatic and electrophoretic forces (vertical) dominated the yeast detachment. The anti-fouling experiment indicates that sufficiently high (-0.5 to -3V) electrical potentials can prevent the attachment of yeast to an electrically charged membrane surface after the initial deposition step.”

Line 319,

“As can be seen from the Fig. S7d, the change of membrane surface charge was not obvious in the presence of emulsified oil foulants and negative electric filed. When the applied voltage reached -3.0V, the membrane surface charge reached -1.16V vs. Ag/AgCl. This led to the smaller interaction force between the Car-PES membrane surface and the emulsified oil foulants, causing severe membrane fouling and rapidly decreasing permeate flux compared with other types of foulants. In addition, Fig. S5d showed that the electronegativity of emulsified oil was high (66.4 mV and 13.9 mV), but by its particle size analysis we can find that most particle size of emulsified oil used in this study were micron level (13.6 μm). It is reasonable to expect that the own gravity of emulsified oil is bigger than applied electric field force and the effect of mitigating membrane fouling was not as obvious as other types of foulants.”

Line 351,

“In this work, the hydrodynamic diameter of BSA (5.4 ± 0.6 nm) is smaller than the Car-PES membrane pore size (65.1 ± 5.8 nm). It indicates that BSA molecules can easily pass through the membrane pore in the absence of electric field.”

Line 353,

“The TOC removal of SA suggest that SA molecules tend to ionized in solution to produce alginate anions with smaller size, so that it can pass through the membrane pores with water flow and deteriorate the SA removal efficiency. In the presence of an electric field, the removal efficiency of SA was increased due to electrostatic repulsion between alginate anions and negative electric field.”

Line 360,

“Oil can be easily deformed, diffused and adhered in aqueous solution. Thus, it is unstable in feed solution and the part of oil molecules may adhere to water molecules and pass through membrane pores.”

Only four typical foulant models are not able to represent wide-spetrum foulants. Please revise the title.

Reply: We totally agree with the reviewer on this point. We have changed the title.

Revision:

Line 2,

“Mitigation of membrane fouling using an electroactive polyether sulfone membrane”

Page S1,

“Mitigation of membrane fouling using an electroactive polyether sulfone membrane”

Section 2.2: what is the thickness of the carbon paper?

Reply: The thickness of carbon paper is 180 μm. We have provided this information.

Revision:

Line 109,

“Then, highly-conductive and porous carbon paper (20 cm × 12 cm × 180 μm) was placed on a glass plate (35 cm × 20 cm) and the dope was cast onto the carbon paper with a casting knife (9 cm × 11 cm × 150 μm).”

Section 2.4: “Duration of the filtration experiments was fixed at 60 min at ambient temperature.” Was the filtration tested for a long time? How was 1 h chosen for the experiment?

Reply: We also performed the filtration experiments as long as 24 h and results suggest that the flux changing trend within 1 h was similar with that of 24 h. We, thus, choose 1 h as a representative. Actually, several reports also adopted 1 h for demonstration, such as Zhang et.al [1], Costa et. al [2] and Luo et. al [3].  

Reference

Zhang, Q.; Vecitis, C.D. Conductive CNT-PVDF membrane for capacitive organic fouling reduction. J. Membr. Sci. 2014, 459, 143-156, doi: https://doi.org/10.1016/j.memsci.2014.02.017. Costa, A.R.; de Pinho, M.N.; Elimelech, M. Mechanisms of colloidal natural organic matter fouling in ultrafiltration. J. Membr. Sci. 2006, 281, 716-725, doi: https://doi.org/10.1016/j.memsci.2006.04.044. Luo, M.-L.; Zhao, J.-Q.; Tang, W.; Pu, C.-S. Hydrophilic modification of poly (ether sulfone) ultrafiltration membrane surface by self-assembly of TiO2 nanoparticles. Appl. Sur. Sci. 2005, 249, 76-84, doi: https://doi.org/10.1016/j.apsusc.2004.11.054.

Revision:

Line 219,

“And over 24 h continuous filtration, the flux changing trend was similar with that of 1 h (Fig. S2).”   Page S3,

Fig. S2. Water fluxes through PES and Car-PES membranes over 24h. (Conditions: [BSA]in=10 mg/L, [SA]in=10 mg/L, [yeast]in=10 mg/L, [emulsified oil]in=10 mg/L, [cross-flow velocity]=6.1 cm/s, [time]=24 h, and [pressure]= 0.1 bar)”

Section 3.1: “pore size compared with the PES (D = 65.1 nm vs. D =77.0 nm).” How was the membrane pore size obtained? 

Reply: The mean pore radius D was obtained by the filtration velocity method.

Revision:

Line 181,

“which obtained by the filtration velocity method (Table S1).”

Page S2,

“Filtration velocity method

According to the Guerout-Elford-Ferry equation, mean pore radius D could be experimentally determined by [1]:

                                                (1)

where η is the water viscosity (8.9 × 10−4 Pažs), ℓ is the membrane thickness (nm), Q is the volume of the permeate water per unit time (m3žs−1), ε is the membrane porosity, A is the membrane effective area (m2) and ΔP is the operational pressure (0.1 MPa).”

References

“[1] Basri, H.; Ismail, A.F.; Aziz, M. Polyethersulfone (PES)–silver composite UF membrane: Effect of silver loading and PVP molecular weight on membrane morphology and antibacterial activity. Desalination 2011, 273, 72-80, doi: https://doi.org/10.1016/j.desal.2010.11.010.”

The pore sizes are much larger than typical thin-film composite (TFC) membranes prepared with MPD and TMC. What is the purpose of the TFC layer? Why the rejections to bovine serum albumin & sodium alginate are so low?

Reply: The main purpose of a TFC layer is to form dense layer on the membrane surface and to improve the membrane selectivity; For the limited rejection for BSA and SA, the reason has been supplemented in the new version.

Revision:

Line 116,

“which formed a dense film on the membrane surface and to improved membrane selectivity.”

Line 351,

“In this work, the hydrodynamic diameter of BSA (5.4 ± 0.6 nm) is smaller than the Car-PES membrane pore size (65.1 ± 5.8 nm). It indicates that BSA molecules can easily pass through the membrane pore in the absence of electric field.”

Line 353,

“The TOC removal of SA suggest that SA molecules tend to ionized in solution to produce alginate anions with smaller size, so that it can pass through the membrane pores with water flow and deteriorate the SA removal efficiency. In the presence of an electric field, the removal efficiency of SA was increased due to electrostatic repulsion between alginate anions and negative electric field.”

“The decrement of the surface roughness of the Car-PES membrane can be explained by the coverage of the carbon paper on the PA layer” why is the membrane roughness lower than the membrane without carbon paper? What is the structure of the membrane (PA layer on the top)? Please also provide the FESEM images of the membrane top surfaces.

Reply: Sorry for the misunderstanding caused. The decrement of the surface roughness of the Car-PES membrane can be explained by the presence of the carbon paper. The pore size of carbon paper was hundreds of micrometers. It is reasonable to expect that the PES polymer could easily infiltrate the pores of the carbon paper during casting, and promote the interface polymerization between the PES layer and PA layer;

The structure of Car-PES membrane has three layers, i.e. the PA layer on the top, the PES layer in the middle, and the carbon paper on the bottom;

We have revised the mechanism section and provided the FESEM images of the membrane top surfaces in the new version.

Revision:

Line 195,

“The pore size of the carbon paper was hundreds of micrometers. It is reasonable to expect that the PES polymer could easily infiltrate the pores of the carbon paper during casting, promoting the interface polymerization between the PES layer and PA layer.”

Line 170,

“The structure of Car-PES membrane has three layers, the PA layer on the top, the PES layer in the middle, and the carbon paper on the bottom.”

Line 182,

“From the Fig. 2e and f, the morphology of the Car-PES top layer was slightly smoother than PES. It can be explained by the addition of carbon paper, promoting the interface polymerization on Car-PES membrane surface.”

Fig. 2. SEM images of the bottom, cross-sections and top of PES (a, b, e) and Car-PES (c, d, f) membranes: (a, c) bottom surfaces, (b, d) cross-sections and (e, f)top surfaces.”

“As is obvious in Fig. S1, the contact angle on the back surface of the Car-PES membrane was only slightly higher than that of PES membrane” why was contact angle on membrane back surface tested? Please also provide the contact angles of membrane top surfaces.

Reply: Sorry for the misunderstanding caused. We have corrected this mistake.

Revision:

Line 199,

“The contact angle on the top surface of the Car-PES membrane was only slightly higher than that of PES membrane, despite the hydrophobic nature of carbon paper.”

Figure 4. What was the original flux of car-PES and the PES membranes? Why the unit of normalized flux is LMH?

Reply: The original flux of Car-PES and the PES membranes was 365 ± 6.3 LMH and 377 ± 5.8LMH, respectively. The normalized flux generally has no units. We have supplemented the original flux in the new version and the unit of normalized flux has been modified.

Revision:

Line 214,

“Fig. 4. Water fluxes through PES and Car-PES membranes. (Conditions: [BSA]in=10 mg/L, [SA]in=10 mg/L, [yeast]in=10 mg/L, [emulsified oil]in=10 mg/L, [cross-flow velocity]=6.1 cm/s, [time]=60 min, and [pressure]= 0.1 bar).”

Line 239,

Fig. 5. Anti-fouling performance as a function of ionic strength. Car-PES cathode and Ti anode: the 1 h normalized permeability vs. aqueous Na2SO4 concentration; (Conditions: [SA]in= 10 mg/L, [voltage]= -1.0 V).”

Line 329,

Fig. 6. Normalized flux through a Car-PES membrane in comprehensive anti-fouling experiments with different foulant solutions and various negative voltages applied: (a) BSA, (b) SA, (c) yeast, (d) emulsified oil. (Conditions: [BSA]in= 10 mg/L, [SA]in= 10 mg/L, [yeast]in= 10 mg/L, [emulsified oil]in= 10 mg/L, [Na2SO4]= 10mM, [cross-flow velocity]= 6.1 cm/s, and [pressure]= 0.1 bar).”

Line 414,

Fig. 9. (a) Normalized fluxex through a Car-PES membrane at different voltages applied. (b) TOC removal efficiency for a Car-PES membrane at different voltages (Conditions: [BSA]in=10 mg/L, [SA]in=10 mg/L, [yeast]in=10 mg/L, [emulsified oil]in=10 mg/L, [Na2SO4]= 10mM, [cross-flow velocity]=6.1 cm/s, and [pressure]=0.1 bar).”

Figure 5. The authors only tested the anti-fouling performance of car-PES. How was the fouling performance of the PES membrane? Was the performance similar to car-PES sample? Will charge influence the anti-fouling performance of the PES membrane? Please justify.

Reply: Thank you for this good suggestion. The fouling performance of the PES membrane was present in Fig. 4. When filtering selected foulants, the anti-fouling performance of both PES and the Car-PES membranes demonstrated a similar trend. Due to the electric insulative nature of the PES membrane, their anti-fouling performance was not affected by the electric field at all. We also performed additional experiments to prove that the applied electric field posed negligible effect to the anti-fouling performance of the PES membrane.

Revision:

Line 220,

“Due to the electric insulative nature of PES alone, the anti-fouling performance of the PES membrane was not affected by the electric field. The one-hour anti-fouling experiment of PES membrane also proved this point (Fig. S3).”

Page S4,

Fig. S3. Normalized flux through a PES membrane in comprehensive anti-fouling experiments with different negative voltages applied (Conditions: [SA]in= 10 mg/L, [Na2SO4]= 10 mM, [cross-flow velocity]= 6.1 cm/s, and [pressure]= 0.1 bar)”

Figure 7. Why do membranes without applied voltage have lower TOC removal efficiencies? Please justify.

Reply: The hydrodynamic diameter of BSA (5.4 ± 0.6 nm) is smaller than the Car-PES membrane pore size (65.1 ± 5.8 nm). It indicates that BSA molecules can easily pass through the membrane pore in the absence of electric field. The TOC removal of SA suggest that SA molecules tend to ionized in solution to produce alginate anions with smaller size, so that it can pass through the membrane pores with water flow and deteriorate the SA removal efficiency. While in the presence of an electric field, the SA removal efficiency was increased due to electrostatic repulsion between alginate anions and the electric field. The TOC removal efficiency of yeast solution remained at high level (>90%), since the yeast particle size is much larger than the pore size of the Car-PES membrane.  Oil can be easily deformed, diffused, and adhered in aqueous solution. Thus, it is unstable in the feed solution and part of oil molecules may adhere to water molecules and pass through membrane pores. We have supplemented related information in the new version.

Revision:

Line 351,

“In this work, the hydrodynamic diameter of BSA (5.4 ± 0.6 nm) is smaller than the Car-PES membrane pore size (65.1 ± 5.8 nm). It indicates that BSA molecules can easily pass through the membrane pore in the absence of electric field.”

Line 353,

“The TOC removal of SA suggest that SA molecules tend to ionized in solution to produce alginate anions with smaller size, so that it can pass through the membrane pores with water flow and deteriorate the SA removal efficiency. In the presence of an electric field, the removal efficiency of SA was increased due to electrostatic repulsion between alginate anions and negative electric field.”

Line 360,

“Oil can be easily deformed, diffused and adhered in aqueous solution. Thus, it is unstable in feed solution and the part of oil molecules may adhere to water molecules and pass through membrane pores.”

Reviewer 3 Report

The review of paper entitled

Mitigation of membrane fouling by wide-spectrum negatively-charged foulants using an electroactive polyether sulfone membrane

The manuscript contain the discussion on the results of the water flux and foulant removal performance of the electroactive membranes with different types of typical negatively-charged model foulants .

The objectives of the study is poorly written, it more like a description of article content and should be improved by providing with wider explanation of the study novelty.

The research materials is rather modest, however in my opinion is sufficient to draw certain conclusions. The data interpretation and scientific discussion is acceptable. However, the language of the manuscript need more thorough inspection to avoid spelling errors such as “impro” in the conclusion part.

All the above mentioned arguments guide me to the conclusion that the manuscript can be publish in present form after minor improvement.

Author Response

Comments by Reviewer #3:

The review of paper entitled Mitigation of membrane fouling by wide-spectrum negatively-charged foulants using an electroactive polyether sulfone membrane. The manuscript contain the discussion on the results of the water flux and foulant removal performance of the electroactive membranes with different types of typical negatively-charged model foulants. The objectives of the study is poorly written, it more like a description of article content and should be improved by providing with wider explanation of the study novelty.

The research materials is rather modest, however in my opinion is sufficient to draw certain conclusions. The data interpretation and scientific discussion is acceptable. However, the language of the manuscript need more thorough inspection to avoid spelling errors such as “impro” in the conclusion part.

All the above mentioned arguments guide me to the conclusion that the manuscript can be publish in present form after minor improvement.

Reply: We appreciate the reviewer’s positive comments. Based on the reviewer’s comments, we have reorganized the introduction section to emphasize the novelty in the new version. Also, the language of the manuscript has been double-checked by native English speaker (one of the co-authors of this revised manuscript—Prof. Wolfgang Sand).

Revision:

Line 4,

“Xiaofeng Fang 1,”

Line 8,

“fxf595@dhu.edu.cn (X.F.);”

Line 17,

“Membrane fouling is the bottleneck limiting the wide application of membrane processes. Herein, we adopted an electroactive polyether sulfone (PES) membrane capable of mitigating fouling by various negatively-charged foulants. To evaluate anti-fouling performance and the underlying mechanism of this electroactive PES membrane, three types of model foulants were selected rationally (e.g., bovine serum albumin (BSA) and sodium alginate (SA) as non-migratory foulants, yeast as proliferative foulant and emulsified oil as spreadable foulant). Water flux and TOC removal efficiency in the filtering process of various foulants were tested under an electric field. Results suggest that under electrochemical assistance, the electroactive PES membrane has an enhanced anti-fouling efficacy. Furthermore, a low electrical field was also effective in mitigating the membrane fouling caused by a mixture of various foulants (containing BSA, SA, yeast and emulsified oil). This result can be attributed to the presence of electrostatic repulsion, which keeps foulants away from the membrane surface. Thereby it hinders the formation of a cake layer and mitigates membrane pore blocking. This work implies that an electrochemical control might provide a promising way to mitigate membrane fouling.”

Line 31,

“fouling mitigation;”

Line 34,

“Great expectations are connected with membrane processes to address the challenging issue of water scarcity.”

Line 52,

“The development of anti-fouling membranes is one of the most promising solutions to this challenging issue.”

Line 58,

“The mechanism of anti-fouling can be explained by different reasons. Both direct effects (direct oxidation/reduction of microbial cells and electrostatic repulsion) and indirect effects (electrochemical H2O2 production, radical generation, pH and temperature changes, electroosmotic flow) have been suggested as possible anti-fouling mechanisms [23]. For example, Zhang et al. [24] attributed the fouling reduction with the application of a negative potential to the increased energy barrier and the decreased collision efficiency of negatively charged organic matter with the membrane surface. In addition, a microscopic amount of H2O2 formed electrochemically near the membrane surface could be lethal to bacteria and inhibit biofilm formation during operation [25].”

Line 67,

“Generally, the membrane materials need to be modified with a conductor to allow for an application of an electric field.”

Line 68,

“Numerous bench-scale studies utilize electrostatic repulsion to repel like-charge foulants.”

Line 74,

“Despite these improvements, most strategies are only capable of dealing with mono-type foulants. However, in the actual sewage are always various types of foulants coexisted. Performance may fail in real conditions for the treatment of industrial wastewater with more complex compositions. Moreover, the synergistic mechanism between the electric field and the membrane needs to be evaluated further for different experimental conditions. In addition, foulant removal efficiency is also another important but often neglected criterion on evaluate membrane performance.

An electroactive membrane operated in cross-flow mode may provide an alternative solution to alleviate the fouling by multiple foulants. The advantages of cross-flow mode include low membrane fouling, flexible operation and the ability to handle high load feed water quality. Together with electrochemistry, the proposed technology offers the prospect of both improving water flux and enhancing foulant removal efficiency. As a proof-of-concept, we first adopted a classical phase inversion method to fabricate conductive PES membranes. Then, we examined the membrane anti-fouling efficacy under various operational conditions using a few common negatively-charged model foulants (e.g., BSA and SA as model non-migratory foulants, yeast as model proliferative foulant, emulsified oil as model spreadable foulant). Finally, we propose an underlying mechanism for fouling mitigation based on our experimental evidence. The outcome of this study is providing a robust and promising strategy for alleviating fouling problems from multiple foulants by integrating state-of-the-art electrochemistry and membrane separation.”

Line 118,

“Characterizations”

Line 426,

“Meanwhile, the TOC removal efficiencies were improved by 91%, 97%, 90% and 94% correspondingly.”

Reviewer 4 Report

The manuscript submitted by Chunyan Ma , Chao Yi , Fang Li , Chensi Shen , Zhiwei Wang , Wolfgang Sand, and Yanbiao Liu elucidates the approach to fouling mitigation by applying an electroactive polyether sulfone membrane. I have the following comments and suggestions:

Please define the terms “non-migratory”, “proliferative” and “spreadable” foulant in the introduction

Line 27 Please use a more scientific term than “a cocktail”

Lines 40-42 the mechanisms of fouling according to Hermia’s pore blocking models and fouling types should be differentiated and explained, not mixed up together in one sentence

Lines 41-42- are “temporary” and “permanent” flux decline the scientific terms? Please provide the references or replace them with “reversible and irreversible fouling”.

Line 44: what other antifouling strategies do you know. Please provide references. The phrase “The development of antifouling membranes is the most fundamental one” did not convince me.

Please explain the mechanisms of action of proper direct current (DC) voltage with regard to membrane fouling mitigation in the introduction

What was the pH of the samples of foulant solutions, during zeta potential measurement? Did you apply pH correction before these measurements?

Fig.2 is not well explained for the reader. Please mark the finger-like cavities in figure 2. Please, additionally provide the figures you refer to (from the sources [19], [23]) as a part of Fig2 in the manuscripts.

Line 167: what do you mean by “partially reflects”?

Lines 195-198: please provide the surface charge of the membranes in the relevant conditions.

Line 219: what do you mean by “external negative voltage”?

One of my greatest concerns with regard to the current manuscript is a quite poor explanation of the underlying mechanisms of fouling mitigation by using the combination of the electrical field and PES membrane in the discussion part. Please, elaborate more on this topic.

Besides, I recommend the authors to improve English since the various terms are not written in the standard scientific English style.

Author Response

Comments by Reviewer #4:

Please define the terms “non-migratory”, “proliferative” and “spreadable” foulant in the introduction

Reply: We have defined these terms in the new version. Thank you for the good suggestion.

Revision:

Line 142,

“Non-migratory foulants refer to organic colloids and NOM, deposit on membrane surfaces and form a rather stable cake layer, we choose BSA and SA as model non-migratory foulants. Proliferative foulants, also referred to as biofoulants, such as bacterial cells, extracellular polymeric substances (EPS) and cell debris, we choose yeast as model proliferative foulant. Spreadable foulants, including various kinds of oils, not only attach, but also spread, coalesce and form a continuous layer on membrane surfaces, we choose emulsified oil as model spreadable foulant.”

Line 27 Please use a more scientific term than “a cocktail”

Reply: We have unified “mixture” to replace “a cocktail throughout the revised manuscript.

Revision:

Line 26,

“Furthermore, a low electrical field was also effective to mitigating the membrane fouling caused by a mixture of various foulants (containing BSA, SA, yeast and emulsified oil).”

Line 392,

“Fouling mitigation with a mixture of various foulants

Line 396,

“comprehensive fouling mitigation of the Car-PES membrane was evaluated by challenging it with a mixture of the several foulants (10 mg/L BSA, 10 mg/L SA, 10 mg/L yeast, 10 mg/L emulsified oil and 10 mM Na2SO4) under different voltages.”

Line 405,

“This confirms that fouling mitigation with a mixture of various foulants is effective. Furthermore, the water flux and TOC removal efficiency of the Car-PES membrane challenged by a mixture of BSA, SA, yeast and emulsified oil, are as good as for a single foulant.”

Line 422,

“Such efficacy has been demonstrated by treating various types of foulants (e.g., BSA, SA yeast and emulsified oil) as well as a mixture of these foulants under electro-assistance.”

Line 428,

“Furthermore, the membrane presented high anti-fouling abilities for removing mixture of various foulants.”

Lines 40-42 the mechanisms of fouling according to Hermia’s pore blocking models and fouling types should be differentiated and explained, not mixed up together in one sentence

Reply: Thank you for the good suggestion, we have revised accordingly.

Revision:

Line 38,

“According to the Hermia’s pore blocking models [3], there are four basic types of fouling, i.e. complete blocking, intermediate blocking, standard blocking and cake layer formation. According to complete blocking model, each molecule that reaches the membrane surface completely blocks the entrance of a membrane pore. The intermediate blocking model is less restrictive, because it considers that some molecules may deposit onto other molecules previously settled. This means that not each molecule, which arrives to the membrane surface, blocks membrane pores. The standard blocking model considers that molecules deposit over the pore walls. As a result, the amount of membrane pores decreases proportionally to the filtered permeate volume. The cake layer formation refers to the formation of a dense layer on the membrane surface.”

References

“[3] Hwang, K.-J.; Lin, T.-T. Effect of morphology of polymeric membrane on the performance of cross-flow microfiltration. J. Membr. Sci. 2002, 199, 41-52, doi: https://doi.org/10.1016/S0376-7388(01)00675-5.”

Lines 41-42- are “temporary” and “permanent” flux decline the scientific terms? Please provide the references or replace them with “reversible and irreversible fouling”

Reply: We have replaced these terms with “reversible” and “irreversible throughout the revised manuscript.

Revision

Line 46,

“All of them can cause reversible and/or irreversible fouling.”

Line 44: what other antifouling strategies do you know. Please provide references. The phrase “The development of antifouling membranes is the most fundamental one” did not convince me.

Reply: We have provided related references on other anti-fouling strategies. Also, we have changed this improper sentence. Thank you!

Revision

Line 49,

“Several methods have been proposed to alleviate membrane fouling in other studies [7], such as pre-treatment of raw water, optimization of operation conditions and membrane cleaning [8-10]. However, the methods can alleviate membrane fouling, but it cannot radically solve the problem and consume space and energy.”

References

“[7] “Zhang, R.; Liu, Y.; He, M.; Su, Y.; Zhao, X.; Elimelech, M.; Jiang, Z. Antifouling membranes for sustainable water purification: strategies and mechanisms. Chem. Soc. Rev. 2016, 45, 5888-5924, doi:10.1039/C5CS00579E”

[8] Gao, W.; Liang, H.; Ma, J.; Han, M.; Chen, Z.-l.; Han, Z.-s.; Li, G.-b. Membrane fouling control in ultrafiltration technology for drinking water production: A review. Desalination 2011, 272, 1-8, doi: https://doi.org/10.1016/j.desal.2011.01.051.

[9] Crozes, G.F.; Jacangelo, J.G.; Anselme, C.; Laîné, J.M. Impact of ultrafiltration operating conditions on membrane irreversible fouling. J. Membr. Sci. 1997, 124, 63-76, doi: https://doi.org/10.1016/S0376-7388(96)00244-X.

[10] Dong, B.-z.; Chen, Y.; Gao, N.-y.; Fan, J.-c. Effect of coagulation pretreatment on the fouling of ultrafiltration membrane. J. Environ. Sci. 2007, 19, 278-283, doi: https://doi.org/10.1016/S1001-0742(07)60045-X.”

Please explain the mechanisms of action of proper direct current (DC) voltage with regard to membrane fouling mitigation in the introduction

Reply: We have supplemented the related information in the new version.

Revision

Line 58,

“The mechanism of anti-fouling can be explained by different reasons. Both direct effects (direct oxidation/reduction of microbial cells and electrostatic repulsion) and indirect effects (electrochemical H2O2 production, radical generation, pH and temperature changes, electroosmotic flow) have been suggested as possible anti-fouling mechanisms [23]. For example, Zhang et al. [24] attributed the fouling reduction with the application of a negative potential to the increased energy barrier and the decreased collision efficiency of negatively charged organic matter with the membrane surface. In addition, a microscopic amount of H2O2 formed electrochemically near the membrane surface could be lethal to bacteria and inhibit biofilm formation during operation [25].”

What was the pH of the samples of foulant solutions, during zeta potential measurement? Did you apply pH correction before these measurements?

Reply: We applied pH correction before the zeta potential measurements. We have mentioned this in the revised manuscript.

Revision

Page S6,

“Fig. S5. Zeta potential: (a) BSA, (b) SA, (c) yeast, (d) emulsified oil. (Conditions: [BSA]in= 10 mg/L, [SA]in= 10 mg/L, [yeast]in= 10 mg/L , [emulsified oil]in= 10 mg/L, PH=7.2). ”

2 is not well explained for the reader. Please mark the finger-like cavities in figure 2. Please, additionally provide the figures you refer to (from the sources [19], [23]) as a part of Fig2 in the manuscripts.

Reply: We have revised the explanations of Fig. 2 according to the reviewer’s comments. 

Revision

Line 185,

Fig. 2. SEM images of the bottom, cross-sections and top of PES (a, b, e) and Car-PES (c, d, f) membranes: (a, c) bottom surfaces, (b, d) cross-sections and (e, f)top surfaces.”

Line 167: what do you mean by “partially reflects”?

Reply: We have corrected this mistake. This setence has been changed to “The decline in water flux indirectly reflects the extent of fouling of the membranes.”

Revision

Line 202,

“The decline in water flux indirectly reflects the extent of fouling of the membranes.”

Lines 195-198: please provide the surface charge of the membranes in the relevant conditions.

Reply: We have supplemented the analysis of the surface charge of the membranes.  

Revision

Line 128,

“Cyclic voltammetry and anode/cathode potential distribution as a function of total cell potential of the membranes were measured by a CHI760E electrochemical workstation (CH Instruments, USA).”

Line 235,

“The primary fouling mitigation mechanism is the potential-induced cathodic surface charge (Fig. S4).”

Page S5,

“Fig. S4. Anode/cathode potential distribution as a function of total cell potential at different ionic strengths: (Conditions: Car-PES memebrane cathode, titanium plate anode, and silver/silver chloride (Ag/AgCl) contrast electrode, [SA]in= 10 mg/L).”

Line 219: what do you mean by “external negative voltage”?

Reply: Sorry for the misunderstanding caused. This refers to a negative voltage. We have corrected this mistake.  

Revision:

Line 258,

“However, the membrane fouling caused by BSA was suppressed mostly in the presence of a negative voltage.”

One of my greatest concerns with regard to the current manuscript is a quite poor explanation of the underlying mechanisms of fouling mitigation by using the combination of the electrical field and PES membrane in the discussion part. Please, elaborate more on this topic.

Reply: Thank you and we have supplemented the related underlying mechanisms in the new version.

Revision:

Line 262,

“The measured currents in the experiments were less than <0.15 Ažm−2 (Fig. S8), indicating Faradaic electron transfer reactions were negligible as expected from the Car-PES membrane. Thus, only electrostatic repulsion effects will be considered in regards to the anti-fouling mechanism. Due to BSA is electronegative, electrostatic repulsion occurred on the Car-PES membrane surface under electric field, thus pushing the BSA molecules away from the membrane surface and reducing membrane fouling.”

Line 283,

“SA could easily ionize in feed solution to provide an abundance of alginate anions, generating the like-charge repulsion between the alginate anions and the applied voltage, thus slowing the decline in water flux. As can be seen from the Fig. S7b, the Car-PES membrane surface charge changed in the presence of SA foulants and negative electric filed simultaneously. When the applied voltage reached -3.0 V, the membrane surface charge reached -1.48 V vs. Ag/AgCl. It is reasonable to expect that the stronger interaction force formed between the Car-PES membrane surface and the SA foulants, causing high permeability flux.”

Line 303,

“Anti-fouling mechanism of yeast foulant can be attributed to the electrostatic repulsion influence of negative electric field. The negative electric field was applied to the Car-PES membrane surface, electrostatic and electrophoretic forces (vertical) dominated the yeast detachment. The anti-fouling experiment indicates that sufficiently high (-0.5 to -3V) electrical potentials can prevent the attachment of yeast to an electrically charged membrane surface after the initial deposition step.”

Line 319,

“As can be seen from the Fig. S7d, the change of membrane surface charge was not obvious in the presence of emulsified oil foulants and negative electric filed. When the applied voltage reached -3.0V, the membrane surface charge reached -1.16V vs. Ag/AgCl. This led to the smaller interaction force between the Car-PES membrane surface and the emulsified oil foulants, causing severe membrane fouling and rapidly decreasing permeate flux compared with other types of foulants. In addition, Fig. S5d showed that the electronegativity of emulsified oil was high (66.4 mV and 13.9 mV), but by its particle size analysis we can find that most particle size of emulsified oil used in this study were micron level (13.6 μm). It is reasonable to expect that the own gravity of emulsified oil is bigger than applied electric field force and the effect of mitigating membrane fouling was not as obvious as other types of foulants.”

Besides, I recommend the authors to improve English since the various terms are not written in the standard scientific English style.

Reply: We have requested our colleague—Prof. Wolfgang Sand—to polish the language. Thank you!

Round 2

Reviewer 2 Report

In the revised manuscript, most of the comments have been addressed by the authors. However, some statements need to be clarified before the manuscript can be published. Acceptance with minor revision is suggested.

Some specific comments are listed below:

What is the purpose of the reference figure in Fig. 2(b)? The polyamide layer is formed by “interfacial polymerization” instead of “interface polymerization”. The authors state that “It can be explained by the addition of carbon paper, promoting the interface polymerization on Car-PES membrane surface.” (line 183). Is there any evidence that a smoother polyamide surface means promoted interfacial polymerization? The statement “The decrement of the surface roughness of the Car-PES membrane can be explained by the presence of the carbon paper on the PA layer” (line 193) is confusing. Based on Fig. 2(d), the carbon paper is not in contact with the PA layer. Please revise the statement. The statement “It is reasonable to expect that the PES polymer could easily infiltrate the pores of the carbon paper during casting, promoting the interface polymerization between the PES layer and PA layer.” (line 195) is questionable. Does PES completely cover the surface of carbon paper? If so, the carbon paper underneath the PES layer may not affect the PA layer much. It is suggested to compare the surface roughness of PES and Car-PES without interfacial polymerization to check the effect of membrane substrates.

Author Response

Our Replies to the Reviewers’ Comments and the Revisions Made to the Manuscript

Comments by Reviewer #2:

In the revised manuscript, most of the comments have been addressed by the authors. However, some statements need to be clarified before the manuscript can be published. Acceptance with minor revision is suggested. Some specific comments are listed below:

Reply: Thank you for all these good suggestions. We have carried out the following point-by-point revisions based on the reviewer’s comments and/or suggestions.

What is the purpose of the reference figure in Fig. 2(b)?

Reply: This is a suggestion from another reviewer. The purpose of the reference figure in Fig. 2b is to make the finger-like cavities clearer.

The polyamide layer is formed by “interfacial polymerization” instead of “interface polymerization”.

Reply: We have corrected this mistake. Thank you! 

Revision:

Line 194,

“The decrease in roughness can be attributed to the interfacial polymerization on the Car-PES and PES membrane surface, which formed a dense film.”

The authors state that “It can be explained by the addition of carbon paper, promoting the interface polymerization on Car-PES membrane surface.” (line 183). Is there any evidence that a smoother polyamide surface means promoted interfacial polymerization?

Reply: Sorry for the misunderstanding caused. We have corrected this improper statement. Thank you!    

Revision:

Line 182,

“As shown from the Fig. 2e and f, as expected, the morphology of the top layer of both membranes were similar, due to the absence of carbon fibers in this layer.”

Fig. 2. SEM images of the bottom, cross-sections and top of PES (a, b, e) and Car-PES (c, d, f) membranes: (a, c) bottom surfaces, (b, d) cross-sections and (e, f) top surfaces.

The statement “The decrement of the surface roughness of the Car-PES membrane can be explained by the presence of the carbon paper on the PA layer” (line 193) is confusing. Based on Fig. 2(d), the carbon paper is not in contact with the PA layer. Please revise the statement.

Reply: Sharp eye! Thank you and we have corrected this mistake.  

Revision:

Line 189,

“As shown in Fig. 3a and b, the mean root squared (Rq) surface roughness of the Car-PES and the PES membrane without PA layer was determined to be 77.1 ± 4.8 nm and 73.5 ± 2.6 nm, respectively. It is reasonable to expect that the carbon paper underneath the PES layer posed negligible effect on the membrane surface. Fig. 3c and d indicated the formation of a PA layer, which reduced the surface roughness for both the Car-PES (41.3 ± 3.6 nm) and the PES (40.1 ± 3.2 nm). The decrease in roughness can be attributed to the interfacial polymerization on the Car-PES and PES membrane surface, leading to the formation of a dense film ontop.”

The statement “It is reasonable to expect that the PES polymer could easily infiltrate the pores of the carbon paper during casting, promoting the interface polymerization between the PES layer and PA layer.” (line 195) is questionable. Does PES completely cover the surface of carbon paper? If so, the carbon paper underneath the PES layer may not affect the PA layer much. It is suggested to compare the surface roughness of PES and Car-PES without interfacial polymerization to check the effect of membrane substrates.

Reply: Thank you for your suggestion. We have compared the surface roughness of PES and Car-PES with and without interfacial polymerization to examine the effect of membrane substrates.

Revision:

Line 189,

“ As shown in Fig. 3a and b, the mean root squared (Rq) surface roughness of the Car-PES and the PES membrane without PA layer was determined to be 77.1 ± 4.8 nm and 73.5 ± 2.6 nm, respectively. It is reasonable to expect that the carbon paper underneath the PES layer posed negligible effect on the membrane surface. Fig. 3c and d indicated the formation of a PA layer, which reduced the surface roughness for both the Car-PES (41.3 ± 3.6 nm) and the PES (40.1 ± 3.2 nm). The decrease in roughness can be attributed to the interfacial polymerization on the Car-PES and PES membrane surface, leading to the formation of a dense film ontop.”

Fig. 3. Surface roughness: (a) PES membrane without PA layer, (b) Car-PES membrane without PA layer, (c) PES membrane with PA layer, (b) Car-PES membrane with PA layer.

Reviewer 4 Report

Dear authors, thank you for the provided revisions. Good luck with your future research!

Author Response

Thank you for your time and good suggestions.